# Hydrogel Emulsion with Encapsulated Safflower Oil Enriched with Açai Extract as a Novel Fat Substitute in Beef Burgers Subjected to Storage in Cold Conditions

**DOI:** 10.3390/molecules27082397

**Published:** 2022-04-07

**Authors:** Monika Hanula, Arkadiusz Szpicer, Elżbieta Górska-Horczyczak, Gohar Khachatryan, Grzegorz Pogorzelski, Ewelina Pogorzelska-Nowicka, Andrzej Poltorak

**Affiliations:** 1Department of Technique and Food Development, Institute of Human Nutrition Sciences, Warsaw University of Life Sciences, Nowoursynowska 159c Street 32, 02-776 Warsaw, Poland; arkadiusz_szpicer@sggw.edu.pl (A.S.); elzbieta_gorska_horczyczak@sggw.edu.pl (E.G.-H.); grzegorz.t.pogorzelski@gmail.com (G.P.); ewelina_pogorzelska@sggw.pl (E.P.-N.); andrzej_poltorak@sggw.edu.pl (A.P.); 2Department of Food Analysis and Evaluation of Food Quality, Faculty of Food Technology, University of Agriculture in Krakow, Mickiewicz Ave. 21, 31-120 Krakow, Poland; gohar.khachatryan@urk.edu.pl

**Keywords:** konjac, linseed flour, fat substitute, volatile compounds, lipid oxidation, encapsulation

## Abstract

This study evaluates the effects of using a fat substitute in beef burgers composed of a hydrogel emulsion enriched with encapsulated safflower oil and açai extract. The influences of the fat substitute on the chemical (TBARS, fatty acids, and volatile compounds profile) and physical (weight loss, cooking loss, water-holding capacity, color, and texture analyses) characteristics of the burgers were analyzed after 0, 4 and 8 days of storage at 4 ± 1 °C. The obtained results were compared with control groups (20 g of tallow or 8 g of safflower oil). The fat substitute used improved burger parameters such as chewiness, hardness and the a* color parameter remained unchanged over storage time. The addition of açai extract slowed the oxidation rate of polyunsaturated fatty acids and reduced the changes in the volatile compounds profile during the storage of burgers. The utilization of a fat substitute enriched the burgers with polyunsaturated fatty acids and lowered the atherogenic index (0.49 raw, 0.58 grilled burger) and the thrombogenicity index (0.8 raw, 1.09 grilled burger), while it increased the hypocholesterolemic/hypercholesterolemic ratio (2.59 raw, 2.09 grilled burger) of consumed meat. Thus, the application of the presented fat substitute in the form of a hydrogel enriched with açai berry extract extended the shelf life of the final product and contributed to the creation of a healthier meat product that met the nutritional recommendations.

## 1. Introduction

The current literature suggests that the development of many human diseases stems from a variety of factors, in which diet type is prominent [1]. Over the last 40 years, the prevalence of obesity has doubled, resulting in ischemic heart disease, strokes, and numerous other diseases. Consumers and international health organizations (the World Health Organization and the European Food Safety Authority) have demanded changes in food quality and content in order to improve the level of human health. Meat products belong to a category of food that is a rich source of saturated and trans fatty acids that can increase the risk of cardiovascular diseases. Due to the evolution of human lifestyles, greater demands for easily accessible and fast-food preparation are observed, for example, beef burgers. Recent studies have focused on meeting these demands through the preparation of burgers with potential health benefits that consider consumers’ needs and the nutritional recommendations of global organizations. This has been achieved through a combination of functional ingredients or by using fat substitutes [1,2,3].

The main challenge of fat substitutes is finding an oil that improves the nutritional profile without affecting consumer acceptability of the final product. To overcome such drawbacks of fat substitutes, many strategies have been proposed such as oil encapsulation and immobilization in oleogels or hydrogels [1]. The conversion of liquid vegetable oils into a solid gel has been the subject of many scientific studies. However, this technique, due to high production cost and the destructive effect of organogelators at high temperatures on the fatty acid profile of the oils, is problematic in the food industry. Unlike oleogels, emulsion hydrogel creation is cheap, simple and requires lower temperatures. Hence, this approach is suitable for heat-sensitive or bioactive compounds that undergo oxidative degradation. Moreover, emulsion hydrogels contain ≤ 50% oil. Thus, they improve the fatty acid profile and also effectively reduce the total fat and caloric content in modified meat products. Furthermore, this method allows for the incorporation of both hydrophilic and hydrophobic functional components into the hydrogel matrix [1]. Researchers have used various emulsifiers, biocomposites, and polysaccharides to create emulsion hydrogels that may be used as fat substitutes. Konjac flour, which is a low-calorie ingredient, has been proposed as a promising biocomposite, as it has been successfully used in dry sausages as a fat substitute in pork [4], salcichón enriched with n-3 [5], and pork liver pates [6].

Current trends in the production of functional foods include the designs of products with lower fat content or with additional health benefits using bioactive compounds and dietary fiber. Açai berries are a popular bioactive additive, due to their antioxidant, antilipidemic, anti-inflammatory, and antiproliferative activities; açai berries have been classified as a ”superfood” [7,8]. In contrast, dietary fibre is one of the most common functional ingredients in foods. The incorporation of fiber into meat can increase the daily intake of dietary fiber with food, which is recommended to be >25 g/day. Clinical and epidemiological studies have shown that dietary fiber can increase the feeling of satiety. Moreover, fiber has been shown to reduce hyperlipidaemia, total cholesterol, and the risk of cancer occurrence, as well as to improve glucose tolerance and gastrointestinal health [9]. Flaxseed flour is another ingredient that possesses health-promoting properties, which contains numerous functional compounds. Flaxseed has a favorable profile of fatty acids (polyunsaturated fatty acids) and is a rich source of dietary fiber, protein, and antioxidants (lignans). Moreover, flaxseed is gluten free and can be used to enrich the diet of people suffering from celiac disease [10].

Therefore, the aim of this study was to develop a healthier burger recipe by using a hydrogel emulsion and to study its effect on chemical and physical parameters of raw and grilled meat during storage. To the best of our knowledge, according to the literature, this type of low-fat burger formulation has not yet been tested.

## 2. Results and Discussion

### 2.1. Effect of Encapsulated Safflower Oil Concentration on the Physical Characteristics of Emulsion Hydrogel

#### 2.1.1. Texture and Rheological Analysis

The values of the hydrogel texture parameters with safflower oil concentrations of 29–48% and a control without oil (0%) are shown in Table 1. The values for the firmness parameter ranged from 5.65 (N) to 11.91 (N). The encapsulated oil content (29%, 33%, and 38%) in the hydrogel emulsion resulted in a statistically significant increase in firmness; a similar trend was observed for the lubricity parameter, where the highest value was 11.22 (N × n). The viscosity and adhesion parameters decreased with increasing oil concentration up to 38%. In contrast, the addition of 42%, 45%, and 48% oil increased these parameters (*p* < 0.05). Hence, the obtained analysis showed that oil concentration influenced the hydrogel texture parameters [11,12].

The analysis of the rheological parameters of the hydrogel with encapsulated safflower oil are shown in Table 2. The analyzed samples all behaved as non-Newtonian, pseudoplastic liquids. Based on the Ostwald–de Waele model fitted to the upper flow curves and n values obtained, it was determined that the addition of oil to the hydrogel emulsion promoted a statistically significant increase in pseudoplasticity. The addition of oil increased the shear stress, which was characterized by a higher shear resistance. This was confirmed by a statistically significant increase in the consistency coefficient (K) value. The highest K value was recorded for the sample with 38% oil (139.42 (Pa×s^n^)). A further increase in oil concentration of the hydrogel decreased the K value. A similar trend was observed for the thixotropy parameter. The report by Cano et al. [13] showed that an increase in the lipid phase content in the studied emulsion increased the K parameter.

#### 2.1.2. SEM and FT-IR Analysis

Figure 1 shows the FT-IR spectra of the analyzed hydrogels with encapsulated safflower oil. The absorbance band characteristics of the oil were recorded for 45% and 0% oil content samples. The spectra showed peaks corresponding to the −CH_3_ groups in the range of 1350–1150 cm^−1^. The C−O bond stretching vibrations belonging to ester groups consisted of two coupled asymmetric vibrations, i.e., C−C(=O)-O and O−C−C, which were observed between 1300 and 1000 cm^−1^. The bands corresponding to C−C(=O) −O vibrations of saturated esters were found between 1240 and 1163 cm^−1^, whereas unsaturated esters were observed at lower wave numbers. The vibrational bands of the O−C−C bonds from esters of primary alcohols were found at 1064−1031 cm^−1^ and from secondary alcohols at 1100 cm^−1^, both types of esters were present in triacylglycerol molecules [14]. The low-intensity band at 1390 cm^−1^ was related to C−H combination vibrations. The bands at 1726 cm^−1^ and 1760 cm^−1^ corresponded to C−H stretching vibrations of the methyl, methylene and ethylene groups. The band at 1725 cm^−1^ corresponded to oleic acid, while saturated and trans-unsaturated triacylglycerols exhibited absorption bands at 1725 cm^−1^ and 1760 cm^−1^. The band at 2145 cm^−1^ was related to C−C and C−H stretching vibrations, and the following vibrations at 2952, 2921, and 2855 cm^−1^ corresponded to the valence −C−H vibrations from −CH_3_ and −CH_2_ groups of triglycerides, respectively [15]. The results indicated that the encapsulation process had no effect on the oil’s structure.

The images of scanning electron microscope SEM are shown in Figure 2 and show that nanocapsules were successfully obtained in a polysaccharide matrix. The size and shape of the capsules varied depending on the oil concentration. The results indicated that, in the case of 29% and 33% oil content samples, the capsule envelope was multi-layered (Figure 2A,B). The size of the capsules decreased with increasing oil concentration (Figure 2C,D). The capsules in the hydrogel emulsion were present throughout the matrix and their distribution was uniform. 

Despite slight statistical differences in the tested variants, the hydrogels with concentrations ranging from 29% to 45% were characterized with similar texture parameters. A 48% oil content was too high and differed significantly from other concentrations. Furthermore, the success of hydrogel applications in food depends largely on the ability of the oleogelator to form a network that traps liquid oil [16]. Therefore, the hydrogel with 45% concentration encapsulated oil was selected for further study as a fat substitute.

#### 2.1.3. Biochemical Analysis of Oil and Açai Extract

The results of the TPC analysis and antioxidant activity of oil and açai extract are shown in Table 3. The concentration of the extract used in the main study was determined based on the study conducted by Mokhtar et al. [17]. Safflower oil containing polyphenols accounted for 0.27 mg gallic acid/g of oil, whereas the antioxidant activity was 0.09 mg ascorbic acid/g of oil according to the ABTS analysis and 0.32 mg ascorbic acid/g of oil according to the FRAP analysis. Nimrouzi et al. [18] reported similar results. 

The fatty acid profile analysis showed that safflower oil was characterized by a high PUFA content, especially n-6. This result was also confirmed by a study conducted by Rutkowska et al. [19].

### 2.2. Main Study

#### 2.2.1. TBARS Analysis

A TBARS analysis is a commonly used indicator of oxidative lipid rancidity in meat, which quantifies the amount of secondary oxidation products. The analysis of the lipid peroxidation content in the studied burgers after 0, 4, and 8 days of storage is shown in Figure 3. The results indicated that, depending on the day of storage and variant, the lipid peroxidation content ranged from 1.14 to 6.92 mg MA/100 g sample. At Day 0, depending on the variant used, the lipid oxidation values ranged from 1.14 (CO) to 5.2 mg MA/100 g sample (GE), hence, the application of a fat substitute in the form of a hydrogel with encapsulated oil and flaxseed flour increased the TBARS values as compared with the control variants (CT and CO). This result was probably due to the presence of flaxseed flour which contained high PUFA levels [20]. The defatted flaxseed flour used in the experiment contained 10 g of fat per 100 g of flour. The manufacturing process of the flaxseed flour could have influenced the generation of free radicals, aldehydes, and ketones upon exposure of PUFAs to light, heat, and oxygen. A study performed by Hautrive et al. [9] revealed that the addition of defatted flaxseed flour elevated TBARS values as compared with a control variant. Furthermore, our results showed that between 0 and 4 days of storage, the highest increase in lipid oxidation (1.85–2.08) was observed in CT and CO. Therefore, using a fat substitute in the form of gel (konjac flour and sodium alginate) with encapsulated oil and flaxseed flour (G, GT, GE, and GET) exerted a positive effect on the reduction in the peroxidation rate. Thus, structuring the added fat significantly reduced the TBARS value (*p* < 0.05, Figure 3) as compared with the product containing animal or vegetable fat [21]. Additionally, our research showed that the MA value increased with storage time, regardless of variant type. As a result, lipid oxidation contributed to the development of rancidity while reducing the quality of meat products during storage [22]. The threshold value that dictates the loss of sensory quality of food is >1.0 mg malondialdehyde/kg burger [23]. In our case, despite the observed increase in fat oxidation, all values obtained were below the threshold value.

#### 2.2.2. Color and pH Analysis

The pH parameter of meat can affect certain characteristics such as color, flavor, aroma, tenderness, and nutritional value. An analysis of the meat pH parameter was carried out on raw burgers after each storage time. Table 4 lists the obtained pH values, which, depending on the variant and storage time, range from 5.56 to 5.81. The addition of a hydrogel and flaxseed flour contributed to the observed increase in pH on Day 0 as compared with the controls (CT and CO). This effect was possibly due to the addition of fiber in the form of flaxseed flour. Hautrive et al. [9] and Sánchez-Zapata et al. [24] also reported higher pH values in modified fiber burgers as compared with control variants. Our study demonstrated that the addition of tallow to the studied meat increased the pH values of the burgers more than the addition of safflower oil (5.64, respectively, 5.56 *p* < 0.05). The observed increase on Day 4 of storage, especially in the G, GT, GE, and GET variants, may have been due to the accumulation of volatile bases such as trimethylamine and ammonia, which formed during protein hydrolysis and amino acid degradation by endogenous enzymes or microorganisms [25].

The effects of hydrogel with encapsulated safflower oil and flaxseed flour on the color parameters of grilled and raw burgers after 0, 4, and 8 days of storage are shown in Table 4. The color parameter of meat influences the willingness of consumers to buy a product due to a growing appreciation for bright red products. The color parameter was significantly affected by storage time and formulation ingredients. The values of the L* parameter in raw burgers increased with increasing storage time regardless of the variant type. A similar trend was observed by Carvalho et al. [26]. However, as compared with the L* color parameter on Day 4, the control variants (CT and CO) had lower values (*p* < 0.05) than the experimental variants (G, GT, GE, and GET). This trend was also observed for a* and b* parameters on Days 4 and 8. Furthermore, the addition of açai extract (GE and GET) had no significant effect on the color parameter regardless of storage time. The fat substitute with microencapsulated oil and flaxseed flour contributed to an improved color parameter of raw burgers after Days 4 and 8 of storage. The analysis of the browning index showed that for both raw and grilled meat, the BI was higher in the G, GT, GE, and GET groups. Probably this effect was caused by the addition of flaxseed flour, which was characterized by a brown color. The color analysis performed on the grilled burgers after each storage day showed no differences among the variants. This was also confirmed by Summo et al. [27] and Gök et al. [28]. However, in the case of the study reported by Lucas-González et al. [29], an increase in the a* color parameter and a decrease in the L* color parameter in grilled burgers with a fat substitute (chestnut flour, emulsion gel, and chia oil) were noted. Grilling changed the color of the meat due to heat-induced denaturation of myoglobin. Currently, the role of fat type used for grilled burger formulation is not fully understood, but it should have less of an effect on the color than other parameters such as storage conditions or pH [27].

#### 2.2.3. TPA Analysis

Texture changes (chewiness, springiness, and hardness) that occur in meat products as a result of replacing animal fat with vegetable fats are becoming an interesting area of research, especially because texture is importance in sensory attributes. The TPA analysis of the grilled burgers is shown in Table 5. The texture parameter values, regardless of variant used and storage time, were significantly (*p* < 0.05) lower (5.3–15.0 (N)) than those of the control samples (24.8–90.9 (N0). This was also observed by Afshari et al. [30], where a fat substitute was applied as an emulsion of soy protein, inulin, β-glucan, canola oil, and olive oil. Similarly, Lucas-González et al. [29] showed that as the degree of fat substitute (chestnut flour emulsion gel and chia oil in pork burgers) increased, texture parameters decreased as compared with control samples. However, Moghtadaei et al. [31] applied a hydrogel (sesame oil and ethyl cellulose) as a fat substitute in a burger and observed the opposite effect as compared with our results. According to the literature, there are many conflicting reports regarding the effect of hydrogel type on the texture parameters of grilled burgers. These differences may occur due to the method of vegetable oil incorporation into the product, its concentration, and the type of hydrogel used [32]. The grilled burgers (G, GT, GE, and GET) had lower hardness values than that of the CT and CO samples. As expected, the grilling process increased hardness of the control samples due to protein denaturation, water loss, and fat loss [33]. In general, the hardness of meat products is related to the size of the fat molecules therein, allowing the formation of an interfacial protein film around the fat globules by salt-soluble proteins (actin, myosin). Thus, an exponential relationship exists between the size of fat globules and the amount of interfacial protein layer [34,35]. In the case of the studied hydrogel, the fat was encapsulated in a polymer blend, whereas in the control samples, an interfacial protein film likely formed and denaturation occurred because of grilling. The variation in hardness between the control samples was probably due to the amount of interfacial protein layer formed, the type of fat used (beef tallow in tissue form, liquid vegetable oil), and its thermal stability. The higher thermal stability of the hydrogel contributed to a greater decrease in hardness of the product after grilling as compared with beef fat or safflower oil. In addition, the values of grilled burger parameters such as springiness and cohesiveness decreased with increasing storage time, whereas the values of chewiness and firmness parameters increased. This effect was related to natural leakage in burgers, which consequently influenced texture parameters [36]. From the consumer’s point of view, reductions in hardness or chewiness parameters are considered to be a favorable characteristics due to their association with enhanced meat quality of the burger [29].

#### 2.2.4. WHC Analysis, Weight Loss, and Cooking Loss Analysis

The food industry demands the appropriate technological properties of a burger related to mass retention after storage and grilling, and therefore, retention of intrinsic water is of key importance. Adequate water retention in meat products not only reduces mass loss, but also causes accumulation of liquid in the package and consequently changes the texture, color, and ultimately, consumer acceptance of the meat product [37]. Figure 4 shows the analysis of water retention in raw and grilled burgers after 0, 4, and 8 days of storage. The values for the variants G, GT, GE, and GET, regardless of storage time, were in the range of 93.5–98.8%. In contrast, the values of the control samples were 31.8–43.2% depending on the period of storage. Our study showed that a fat substitute with flaxseed flour (G, GT, GE, and GET) improved the WHC about 54–60% as compared with those of the control samples, depending on the storage period and variant. A similar trend was observed by Zinina et al. [38], who observed an increased WHC parameter when flaxseed flour was used. In contrast, a study by Sharefiabadi et al. [39] revealed that the addition of flaxseed flour and coconut flour did not improve the WHC parameter in chicken pasties. The difference in the aforementioned studies may have been the type of fat substitute added to the meat products.

The analysis of mass loss conducted in this study showed that mass loss increased with storage time (*p* < 0.05) in the burgers with a fat substitute as well as the control variants. Nevertheless, the G, GT, GE, and GET variants had much lower mass loss (0.5–0.6%) than that of the CT and CO variants (1.3–2.6%). The replacement of fat with a hydrogel (konjac flour and sodium alginate) enriched with encapsulated oil with flaxseed flour helped to reduce mass changes during storage. The significant difference (*p* < 0.05) observed between the tallow and liquid oil controls was possibly due to the form of added fat [40]. Similar trends were observed for grilled samples after 0, 4, and 8 days of storage, where the fat substitute based on konjac flour and sodium alginate enriched with encapsulated safflower oil showed statistically significant (*p* < 0.05) reduction in the percentage mass loss of the burger. A study by Salcedo-Sandaval et al. [41] showed that konjac gel with added oils (olive oil, flaxseed oil, and fish oil) showed lower mass loss after grilling and baking as compared with control variants. The limiting role in mass changes during grilling was also supported by Moghtadaei et al. [35]. The analysis showed that the addition of tallow (GT and GET) increased the percentage mass loss of the grilled burgers, which could be due to changes in the consistency of the tallow under the influence of the grilling process.

#### 2.2.5. Analysis of the Volatile Compounds Profile

Fat is a precursor to many aromatic compounds, and it also acts as a solvent for these compounds. Changes in the amount and composition of fat can affect the rate and type of processes that occur during burger storage [1]. The analysis of changes to the profile of volatile compounds in raw and grilled burgers after 0, 4, and 8 days of storage is shown in Figure 5. The obtained results displayed differences in the volatile compounds profile between G and GT, between G and GE, between GET and CT, and between GET and CO variants on Day 0. The data revealed that the presence of the extract affected the aroma profile of the raw burger by reducing the changes in compound growth (Figure 5A), thus, slowing down changes that naturally occur in the burger during storage. The alterations in the volatile compounds profile of burgers with a fat substitute after 4 and 8 days of storage were comparable. The opposite was observed for burgers with tallow and oil, where changes increased with storage time, and the process of changes progressed fastest between 4 and 8 days of storage. In contrast, for grilled burgers (Figure 5B), the variants used showed no difference in the volatile compounds profile regardless of storage time, with only a slight variation at Day 0 in GET. However, this may stem from the presence of both extract and tallow. Fluctuations in volatile compounds profile for the CT variant changed dramatically at Day 8, which was possibly due to the progressive oxidative processes of high tallow content (20%), especially after 4 days of storage. In contrast, the CT variant with safflower oil displayed linear changes, which indicated gradual changes that progressed with storage time. 

Table 6 shows the component analysis of the volatile compounds profile of raw burgers with a fat substitute (G, GT, GE, and GET) and control variants (CT and CO) after 0, 4, and 8 days of storage. The analysis revealed that the number of compounds, especially corresponding to aldehydes, alcohols, esters, and ketones, increased with storage time. The addition of oils with increased amounts of unsaturated fatty acids promotes oxidative processes in meat and consequently increases the content of volatile compounds formed by lipid oxidation [26]. Volatile compounds such as alcohols, aldehydes, and ketones are mainly responsible for reducing the sensory quality of meat products [42]. Wantanabe et al. [43] confirmed our findings, where the number of volatile compounds related to alcohols increased with storage time. In contrast, alcohols such as 1−hexanol, 1−pentanol, and 1−octen−3−ol have been considered to be key indicators of lipid oxidation in meat [26]. The component analysis of the volatile compounds profile in grilled burgers was dominated by aldehyde and alcohol group bearing compounds. Emerging compounds such as benzaldehyde, 3−methylbutanal, 2−methylpropanal, pyrazine, and dimethyl sulphide are typical compounds that arise from the Maillard reaction, which is induced by a thermal process [44,45]. The obtained results showed that, apart from lipid degradation and the Maillard reaction, the applied product components were the main factors that influenced the formation of volatile substance characteristics for a particular variant, for example, compounds such as benzaneacetaldehyde, 1−propanol−2 methyl, and pyrazine for variants G, GT, GE, and GET; pentan−2−ol for GE and GET; 2−furanmethanol for CO; and dimethyl sulphide for CT.

#### 2.2.6. Analysis of Fatty Acid Profile and Nutritional Indexes AI, TI, and h/H

The analysis of the fatty acid profile and nutritional indexes of raw and grilled burgers after 0, 4, and 8 days of storage are shown in Table 7. The fatty acid profile was influenced by the type of formulation used. The analysis of saturated fatty acids (SFA) and unsaturated fatty acids (MUFA and PUFA) in raw burgers showed that SFA increased and MUFA and PUFA decreased with storage time, regardless of the variant used. Our results were consistent with the theory that as the number of double bonds in a fatty acid increased, the susceptibility to oxidation increased [26]. As predicted, the addition of safflower oil significantly increased the PUFA content of the raw burgers from 1% (CT) to 24% (GET). The analysis of PUFAs and MUFAs in reformulated raw burgers showed that the addition of açai extract (GE) contributed to lower oxidation levels. Carvalho et al. [26] and Rutkowska et al. [46] confirmed a positive effect of extracts (pitanga leaves, guarana seeds, and chokeberry fruit) on reducing the rate of oxidative changes in fat. Safflower oil used in the experiment was characterized by high PUFA content (79%), with an n − 6/n − 3 ratio of 526. In contrast, the applied reformulation of the burger with cassava oil with the addition of flaxseed flour reduced the n − 6/n − 3 ratio by as much as 10-fold (50 for CO and 5 for GE), while maintaining high PUFA values. According to dietary recommendations, this ratio should be 5-4:1 [19]. For other nutritional indexes, it is recommended that meat products should have the lowest possible AI and TI values, while the h/H ratio should be as high as possible [47]. In our study, independent of storage time, the burger variants with the fat substitute used had the lowest AI and TI values as compared with CT. Furthermore, hydrogel emulsion with flaxseed flour significantly increased the h/H ratio from 1.3 (CT) to 2.5 (GE). Considering the indexes of healthiness that were analyzed, the best results were obtained for raw burgers with the addition of açai extract (GE) in relation to hydrogel without extract (G). The correlation analysis among SFA, MUFA, PUFA, and CLA and storage day, showed that SFA had a very high positive correlation (0.736, *p* < 0.05). In contrast, the correlation analysis of MUFA and CLA showed moderately high negative correlations (−0.359 and −0.488, *p* < 0.05, respectively). In contrast, the PUFA analysis showed no correlation with storage time. The correlations obtained indicated that changes occurred during storage related to the fat oxidation process as a result of which, polyunsaturated acids were converted to a saturated form [26]. The analysis of the fatty acid profile of grilled burgers showed that the use of reformulated burgers contributed to an increase in PUFA content as compared with CT and CO. In addition, usage of hydrogels contributed to lowering n − 6/n − 3 acid ratios, from 24 (CO) to 5 (GE), which was extremely important to maintain proper nutritional standards. Furthermore, the use of reformulation resulted in improved health values of the grilled burger by lowering the Al and Tl values and increasing the h/H ratio.

#### 2.2.7. Analysis Correlation Coefficients among Color Parameters (L*, a*, b*, BI), Textural (Springiness, Chewiness, Cohesiveness, and Hardness), Cooking Loss, Weight Loss, Fatty Acid Profile (SFA, MUFA, and PUFA), WHC, pH, and TBARS in Raw and Grilled Burgers

The correlation analysis among fatty acid profiles (SFA, MUFA, and PUFA), color parameters (L*, a*, b*, and BI), WHC, pH, weight loss, and TBARS in raw burgers is presented in Table 8. The analysis showed a high negative correlation between PUFA and SFA and between PUFA and MUFA (−0.580 and −0.790, *p* < 0.05, respectively). These correlation are probably related to the change in the number of bonds in fatty acids associated with oxidation processes. In addition, the amount of SFA in the raw burger is negatively correlated with the color parameter a* and BI; this may indicate an oxidative process during which the color of the fat changes. The analysis of the burger brightness parameter (L*) showed positive correlations with b*, WHC, pH, TBARS (0.727, 0.539, 0.497, and 0.848, *p* < 0.05, respectively). Moreover, the correlation analysis showed high positive correlations of the parameter b* with BI, WHC, pH, and the TBARS analyses (0.751, 0.928, 0.880, and 0.765, *p* < 0.05, respectively). The correlation analysis among fatty acid profiles (SFA, MUFA, and PUFA), color parameters (L*, a*, b*, and BI), WHC, pH, weight loss, and TBARS in raw burgers is presented in Table 8. The analysis showed a high negative correlation between PUFA and SFA and between PUFA and MUFA (−0.580 and −0.790, *p* < 0.05, respectively). This correlation is probably related to the change in the number of bonds in fatty acids associated with oxidation processes. In addition, the amount of SFA in the raw burger was negatively correlated with the color parameter a* and BI; this may indicate an oxidative process during which the color of the fat changes. The analysis of the burger brightness parameter (L*) showed positive correlations with b*, WHC, pH, and TBARS (0.727, 0.539, 0.497, and 0.848, *p* < 0.05, respectively). Moreover, the correlation analysis showed high positive correlations of the parameter b* with BI, WHC, pH, and the TBARS analyses (0.751, 0.928, 0.880, and 0.765, *p* < 0.05, respectively).

## 3. Materials and Methods

### 3.1. Preliminary Studies

Preliminary studies were conducted to verify the maximum amount of oil absorbed by the base hydrogel without affecting the gel structure. The base for gel formulation consisted of konjac flour (green essence) and sodium alginate (Agnex 1999). Preparation of the gel with encapsulated safflower oil (gold press) was conducted in a multistep process. Gelation involved dissolving 2% sodium alginate and konjac flour in H_2_O at 60 °C with constant stirring until a clear emulsion was obtained. An emulsion was obtained of the mixture of safflower oil and water at a ratio of 2:1. Next, homogenization of the emulsion in base gel was performed using an immersion homogenizer (IKA T18 digital, Ultra Turrax, Germany). The concentrations of encapsulated oil in the emulsion hydrogels were 0% (control), 29%, 33%, 38%, 42%, 45%, and 48%. The biocomposites obtained through this approach were used for further studies: SEM image analysis, FT-IR spectroscopy analysis, texture, and rheology analysis, from which the optimal hydrogel was selected for further analysis and application (see Results and Discussion section).

### 3.2. Beef Burgers Formation and Packaging

The beef (neck muscles, Zakłady Mięsne Wierzejki, Poland) without excess fat and connective tissue was minced. Then, the beef and beef fat (tallow) were ground separately using a meat grinder with an 8 mm hole diameter plate (PI-22-TU-T, Edesa, Greece). Table 9 shows the composition of the prepared burgers with encapsulated oil without/with addition of açai extract (superfoods, PL-EKO-07) in hydrogel and defatted flaxseed flour (LenVitol, Oleofarm, Wrocław, Poland) (variants G, GT, GE, and GET). An emulsion hydrogel containing 45% encapsulated oil was prepared according to the recipe presented in the preliminary studies. The control variant was a burger with tallow (CT) as the conventional source of fat in meat products and oil (CO) as an alternative source of fat. All ingredients, according to the specified proportions, were mixed and burgers (120 g ± 1) of 10 cm diameter were formed. Then, the burgers were placed on 137 × 187 × 50 mm trays made from polyethylene terephthalate (PET) with an absorbent pad (absorbency 1700 mL/m^2^). The trays were then sealed with 35 µm thick PSF films (PSF35ZAC, PolTechPack, Olstzn, Poland). Burgers were packed in modified atmosphere (80% O_2_ and 20% CO_2_) and stored for 0, 4, and 8 days at a temperature of 4 ± 1 °C. Color, pH, fatty acid profile, thiobarbituric acid reactive substances (TBARS), weight loss, water-holding capacity (WHC), and volatile compounds profile analyses were performed on the raw burgers. After the specified storage time (0, 4, and 8 days), the burgers were grilled on an electric grill (silex), heated to 190 °C (bottom plate) and 210 °C (top plate) until 75 °C at the geometric center of each burger (measured with a thermocouple), and then cooled to 25 °C. The prepared burgers were used for the following analyses: color, fatty acid profile, cooking loss, volatile compounds and texture profile analysis (TPA) test. The experimental set-up covered 3 biological replicates (162 samples).

### 3.3. Rheology Analysis

Melt flow curves were measured using a MARS III rheometer (MTMC—MARS Temperature Module) in CR mode with a coaxial cylinder measuring system (CC25 DIN Ti) with a 5.3 mm gap, wherein the measurements were determined at 5 °C. The shear rate was increased from 0 to 80 s^−1^, for 5 min, and then decreased from 80 to 0 s^−1^, for 5 min. The Ostwald–de Waele rheological analysis (Steffe 1996) was used to describe the melt flow curves. Thixotropy and hysteresis loop areas were calculated according to the method described by Sikora et al. [48].

### 3.4. Analysis of the Textural Parameters of Hydrogel Emulsion

Analysis of the hydrogel emulsion and burger texture textural parameters of the hydrogel were analyzed using a texture analyzer (TA.XTplusC Texture Analyzer) by penetration using Perspex 45° TTC Spreadability Rig Cone Sensors (Stable Micro System, Ltd., Goldaming, UK), according to method described by Öğütcü and Yilmaz [12] with some modifications. Initially, the temperature of the sample was maintained at 5 °C for 24 h, then, penetration was carried out to a depth of 23 mm, at a speed of 3 mm/s. From the data obtained, the following parameters were determined using exponent connect software: firmness (N), lubricity (N × s), viscosity (Pa s) and stickiness (N × s).

The texture profile analysis (TPA) of the grilled beef burgers after 0, 4, and 8 days of storage at 4 ± 1 °C was conducted in a double compression cycle using an Instron 5965 universal testing machine (Instron, USA) with a 500 (N) load cell connected to the Bluehill^®^2 software, following the methodology described by Afshari et al. [13]. Samples of 2.45 cm in diameter and 2.50 cm height, cut out from the center of the burger, were subjected to double compression with 3 s relaxation time until their initial height was reduced by 50%. The analysis was performed at a constant head travel speed of 200 mm∙min^−1^, at a temperature of 4 ± 1 °C. Six measurements were made for each group. Texture parameters, such as cohesiveness (−) (the ratio of the area under the curve from the second compression to the area under the curve from the first compression), hardness (N0 (the maximum force of the first compression), elasticity (−), and chewiness (N) (hardness × cohesiveness × elasticity) were calculated using methodology described by Półtorak et al. [49]. 

### 3.5. Scanning Electron Microscope (SEM) Analysis

The morphology of the prepared oil-encapsulated gels was examined using a JEOL JSM-7500F high-resolution scanning electron microscope.

### 3.6. Fourier Transform Infrared Spectroscopy (FT-IR) Analysis

The FT-IR spectra of the biocomposites were recorded in the range of 4000–700 cm^−1^ using a MATTSON 3000 FT-IR spectrophotometer (Madison, Wisconsin, USA). The instrument was equipped with a 30SPEC 30 Degree Reflectance adapter and a MIRacle ATR accessory (PIKE Technologies Inc., Madison, WI, USA). The FT-IR spectra were performed on dried gels (films were obtained).

### 3.7. Water-Holding Capacity (WHC) Analysis

The WHC of raw burgers after 0, 4 and 8 days of storage was analyzed according to the method reported by Grau and Hamm [50]. Three hundred milligrams of the sample was placed on Whatman No. 1 filter paper between two glass plates under a weight of 2000 g for 5 min. In order to take press stain images, a Kaiser system (Germany) equipped with the OImaging MicroPublisher 5.0 RTV software (Canada) was used. Meat and fluid areas were evaluated using the Image-Pro Plus software (v.7.0). The WHC values were calculated using the following formula: WHC% = (Am)/Ap × 100(1)

Am—meat areas; Ap—fluid areas.

### 3.8. Color and pH Analysis

The color evaluation was performed on raw and grilled burgers after 0, 4, and 8 days of storage. Measurements were obtained at 5 different locations on the burgers’ surfaces. The analysis was performed in a CIE L*a*b* system using a Minolta CR-400 chromameter with a CA-A98 attachment (Konica Minolta Inc., Tokyo, Japan) and D65 illuminant. The browning index (BI) was calculated using L*, a*, and b* values according to the following formula [51]:BI = 100 (x − 0.31)/0.17(2)
where x = (a* + 1.75 L*)/(5.645 L* + a* − 3.012 b*).

The pH was measured using a digital pH meter (FiveEasy F20, Mettler Toledo, Warsaw, Poland).

### 3.9. Weight Loss and Cooking Loss during Storage Analysis

The technological properties of the tested burgers were determined in 3 biological replicates using 2 samples for each variant. Raw burgers were weighed on Day 0 and after each day of storage to determine mass loss during storage. Samples were also weighed after grilling and cooling the burgers to room temperature (25 °C). Mass loss after both storage (F) and grilling (G) were calculated according to the following equations:weight loss% = (BR0 − BRx)/BR0 × 100(3)
cooking loss% = (BRx − BGx)/BRx × 100(4)

BR0—raw burger weight on day 0; BRx—raw burger weight after each day of storage; BGx—grilled burger weight after each day of storage

### 3.10. Analysis of Total Phenolic Content (TPC) and Antioxidant Activity of the 2,2-Azinobis(3-ethylbenzothiazoline-6-suslfonic Acid (ABTS) and Ferric Reducing Antioxidant Power (FRAP)

#### 3.10.1. Extraction Process

Safflower oil extraction for the analysis of TPC and antioxidant activity was prepared according to the procedure described by Ablay et al. [52]. Briefly, 5 g of oil was shaken in 5 mL of n-hexane for 5 min. Then, 5 mL of MeOH/H_2_O (80:20, *v*/*v*) was added, centrifuged, and the resulting extract was stored at 4 °C. TPC and antioxidant activity of the açai extract (superfoods, PL-EKO-07) was analyzed after extraction according to the study reported by Hanula et al. [8] with minor modifications. First, 1 g of lyophilized extract was shaken in 25 mL of H_2_O for 1 h. Then, the extraction process was carried out using ultrasound for 5 min. The obtained extract was centrifuged and stored at 4 °C for further analysis. 

#### 3.10.2. TPC Analysis

The TPC analysis of the studied oil and açai extract was carried out using the method described by Singleton and Rossi [53] with some modifications. First, 0.1 mL of extract, 0.5 mL of Folin–Ciocalteu, 2.9 mL of H_2_O, and 1.5 mL of 7% Na_2_CO_3_ were mixed and incubated for 40 min in the dark. The absorbance was measured at 765 nm using a UV-Vis spectrophotometer. The obtained results were expressed as mg gallic acid/g sample.

#### 3.10.3. ABTS and FRAP Analyses

The ABTS and FRAP analyses were performed according to the methods described by Belwal et al. [54] with slight modifications. The ABTS analysis was conducted mixing 0.1 mL of extract with 2.9 mL of ABTS and incubation for 30 min. The FRAP analysis was based on mixing 2.9 mL of FRAP solution (20 mM ferric chloride in H_2_O, 10 mM 2,4,6-tri(2-pyridyl)-s-triazine in 40 mM hydrochloric acid, and 300 mM sodium acetate buffer at a 1:1:10 ratio) with 0.1 mL of the extract and further incubation in the dark for 15 min. The ABTS and FRAP concentrations were measured at 520 nm and 593 nm, respectively, using a UV-Vis spectrophotometer. The results were expressed as mg ascorbic acid/g sample.

### 3.11. Thiobarbituric Acid Reactive Substances (TBARS) Analysis

The analysis of lipid oxidation was evaluated by TBARS changes and was performed according to the procedure described by Brodowska et al. [55]. 1,1,3,3-Tetramethoxypropane was used to prepare the standard curve. The absorbance of the resulting color complex was measured using a UV-Vis spectrophotometer (UV-1800, Shimadzu Corp., 115 VAC, Tokyo, Japan). The TBARS values were calculated in mg of malondialdehyde (MDA) per 100 g sample.

### 3.12. Fatty Acid Profile Analysis, Thrombogenicity Index (Tl), Atherogenic Index (AI), and Hypocholesterolemic/Hypercholesterolemic (h/H) Ratio Analysis

The fatty acid profile analysis was performed for safflower oil in each burger variant (raw and grilled) after 0, 4, and 8 days of storage. Lipids were directly methylated as described by Wojtasik-Kalinowska et al. [56] and Heck et al. [3] with slight modifications. The fatty acid methyl ester composition was analyzed using gas chromatography (Shimadzu GC-2010) with a flame ionization detector (FID) equipped with a Zebron ZB-FAME column (GC Cap. Column, 60 mL × 0.25 mm ID × 0.2 µm df). The initial column temperature was 100 °C held for 3 min, which was increased to 240 °C at a rate of 2.5 °C/min, and held for 10 min. The detector was maintained at 260 °C. In order to identify the FAME composition in burgers or oils, FAME Mix-37 standard (Supelco, TraceCERT^®^, EC:200-838-9, SKU: CRM47885) was used. The obtained results were presented as a fatty acid profile. In addition, TI and AI were determined according to the method described by Ulbricht and Southgate [57] and the hypocholesterolemic/hypercholesterolemic (h/H) ratio was calculated according to Fernandez et al. [58] as follows:TI = (C14:0 + C16:0 + 18:0)/((0.5 × ΣMUFA) + (0.5 × Σn − 6) + (3 × Σn−3) + ((Σn − 3)/(Σn − 6)))(5)
AI = (C12:0 + (4 × C14:0) + 16:0)/((ΣPUFA n − 3) + (ΣPUFA n − 6) + (ΣMUFA))(6)
h/H = (C18:1n9 + ΣPUFA)/(C14:0 + C16:0)(7)

### 3.13. Analysis of the Volatile Compounds Profile

The analysis of the volatile compounds profile was performed using a Heracles II e-nose (Alpha MOS Co., Toulouse, France)based on ultrafast gas chromatograph with a flame ionization detector and a retention index counting application using the AroChemBase library (AlphaSoft software, Alpha MOS Co., Toulouse, France). The gas chromatograph was equipped with two capillary columns of different polarities, i.e., DB-5 and DB-1701, with 10 m × 0.18 mm ID x 0.4 μm film thickness. The analysis was conducted according to the methodology reported by Górska-Horczyczak et al. [59]. Calibration was performed on a standard mixture of C6-C16 alkanes (Restek, ANCHEM Plus, Warsaw, Poland).

### 3.14. Statistical Analysis

The results were analyzed using the Statistica software version 13.3 (StatSoft, Tulsa, OK, USA). The normality of data distribution was verified using the Shapiro–Wilk test. Factorial analysis of variance (ANOVA) was performed in the case of the TBARS analysis. The results of hydrogel texture and rheology as well as color of meat were analyzed with one-way ANOVA. Texture parameters of the burgers were subjected to the Kruskal–Wallis ANOVA, followed by multiple comparisons of mean ranks. Moreover, the strengths of the relationships among fatty acid profile (SFA, MUFA, and PUFA), texture parameters (springiness, chewiness, cohesiveness, and hardness), cooking loss, weight loss, WHC, pH, and color parameters (L*, a*, b*, and BI) methods were determined using Pearson’s correlation coefficients. For all analyses, 95% confidence intervals were established. The AlfaSoft package with statistical quality control was used in order to perform a comparative analysis and to evaluate the chromatographic fingerprints of the volatile compounds.

## 4. Conclusions

In conclusion, an emulsion hydrogel formulated based on konjac flour and sodium alginate with encapsulated oil was used as a functional ingredient in beef burgers enriched with flaxseed flour to develop a healthier alternative. Each variant of the fat substitute containing extract contributed to a reduction in the changes of volatile compounds profile rate and preserved a* color parameter. In addition, the fat substitute reduced the hardness and chewiness parameters of the grilled burgers. According to the obtained results, the applied fat substitute promoted a reduction in the nutritional AI and TI indexes, as well as increased the amount of the PUFAs and h/H ratio. Thus, a hydrogel emulsion enriched with encapsulated oil and açai extract along with the addition of flaxseed flour in a burger recipe instead of beef fat results in a healthier alternative that is rich in fiber.

## Figures and Tables

**Figure 1 molecules-27-02397-f001:**
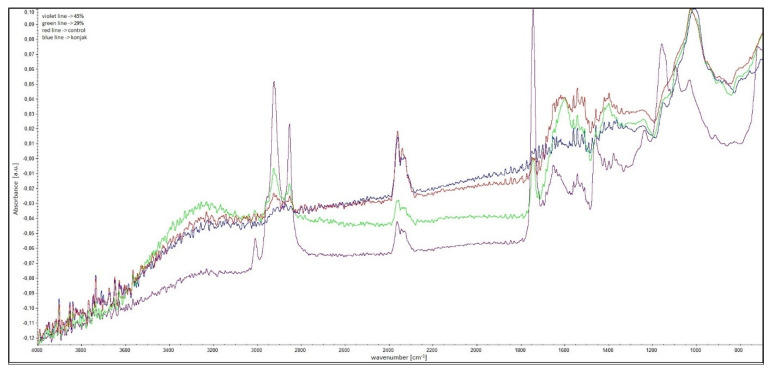
FT-IR analysis of hydrogels with encapsulated safflower oil.

**Figure 2 molecules-27-02397-f002:**
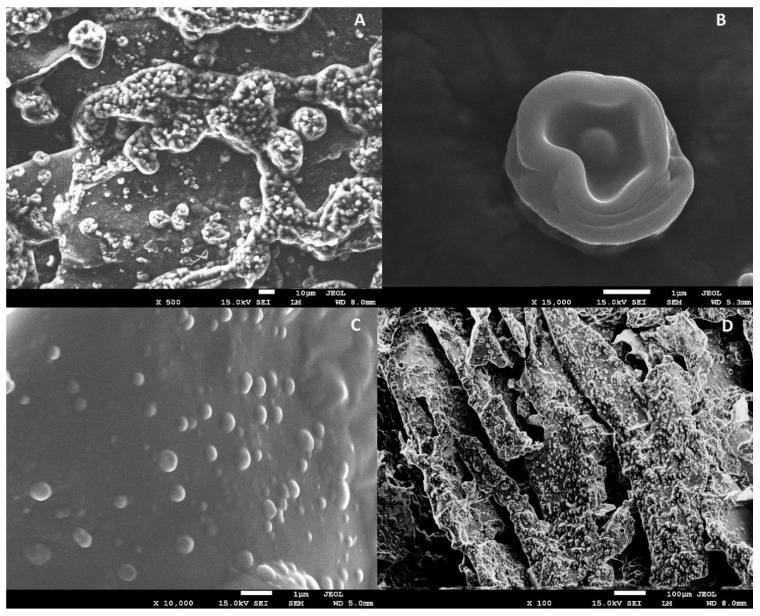
SEM analysis of hydrogels with encapsulated oil: (**A**,**B**) Concentration of 29%; (**C**,**D**) concentration in the range of 45%.

**Figure 3 molecules-27-02397-f003:**
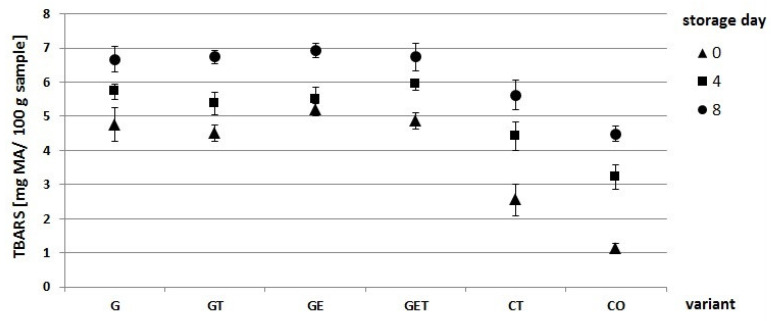
Analysis of lipid oxidation (TBARS) in burgers with a fat substitute after 0, 4, and 8 days of storage. G—encapsulated oil; GT—encapsulated oil + tallow; GE—encapsulated oil with açai extract; GET—encapsulated oil with açai extract + tallow; CT—control with tallow; CO—control with oil.

**Figure 4 molecules-27-02397-f004:**
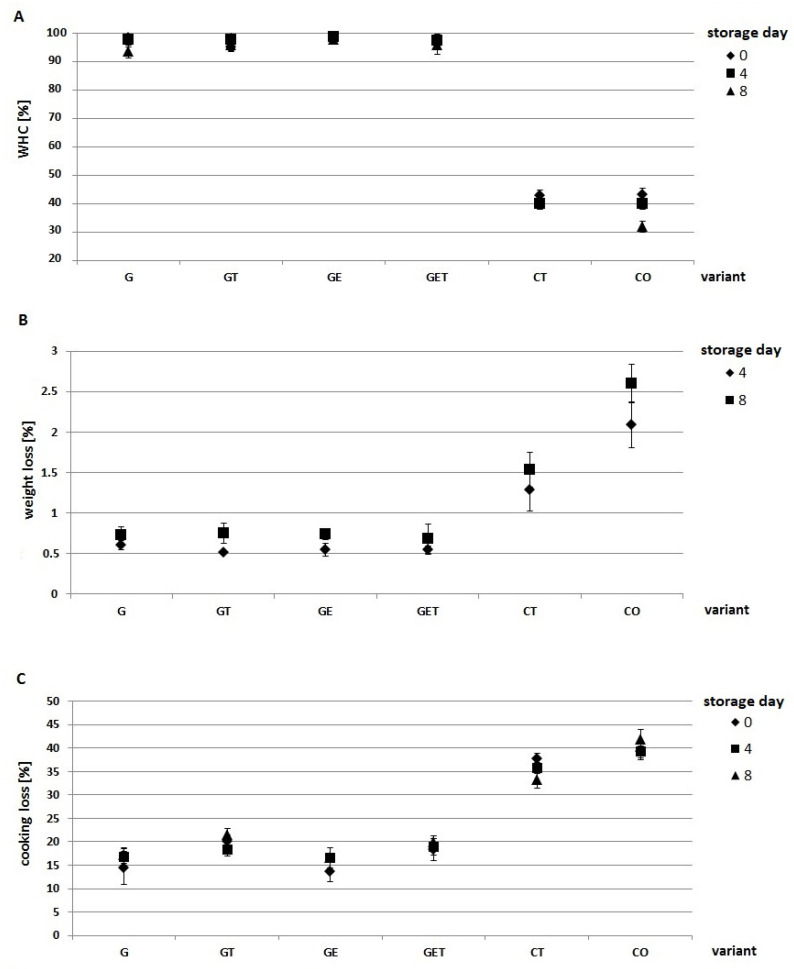
Effect of a fat substitute on water retention (WHC) (**A**); weight loss (**B**); and cooking loss (**C**), after 0, 4, and 8 days of storage. G—encapsulated oil; GT—encapsulated oil + tallow; GE—encapsulated oil with açai extract; GET—encapsulated oil with açai extract + tallow; CT—control with tallow; CO—control with oil.

**Figure 5 molecules-27-02397-f005:**
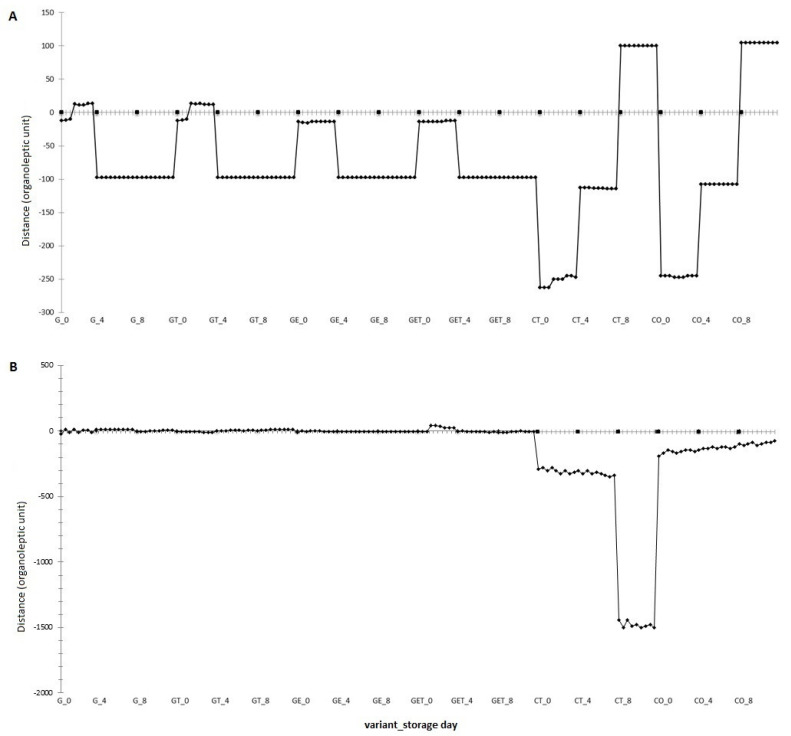
Changes in the volatile compounds profile of raw burgers (**A**); grilled burgers (**B**), after 0, 4, and 8 days of storage (4 ± 1 °C). G—encapsulated oil; GT—encapsulated oil + tallow; GE—encapsulated oil with açai extract; GET—encapsulated oil with açai extract + tallow; CT—control with tallow; CO—control with oil.

**Table 1 molecules-27-02397-t001:** Effect of oil concentration on the hydrogel texture parameters.

Oil (%)	Firmness (N)	Lubricity (N × s)	Viscosity (N)	Adhesiveness (N × s)
0	5.65 ± 0.34 ^A^	4.81 ± 0.39 ^A^	−3.38 ± 0.21 ^D^	−1.02 ± 0.04 ^E^
29	10.15 ± 0.44 ^C^	9.41 ± 0.55 ^C,D^	−6.01 ± 0.59 ^C^	−2.10 ± 0.25 ^B,C^
33	11.17 ± 0.69 ^D^	10.37 ± 1.02 ^D,E^	−7.03 ± 0.56 ^B^	−2.42 ± 0.29 ^B^
38	11.91 ± 0.43 ^D^	11.22 ± 0.61 ^E^	−8.03 ± 0.30 ^A^	−2.45 ± 0.32 ^A^
42	10.20 ± 1.03 ^C^	9.17 ± 1.04 ^C^	−7.10 ± 0.69 ^B^	−1.90 ± 0.35 ^C,D^
45	9.31 ± 0.61 ^B,C^	8.66 ± 0.90 ^B,C^	−6.61 ± 0.24 ^B,C^	−1.87 ± 0.41 ^C,D^
48	8.89 ± 1.16 ^B^	8.07 ± 1.10 ^B^	−6.37 ± 0.82 ^C^	−1.47 ± 0.66 ^D^

^A–E^—Mean values between variants in the same column indicated by different letters indicate a statistically significant difference.

**Table 2 molecules-27-02397-t002:** Parameters of the Ostwald–de Waele rheological model and area of the hysteresis loop of the hydrogel with encapsulated safflower oil.

Oil (%)	Ostwald–de Waele Model			Area of Hysteresis Loop (Pa × s)	
	K (Pa × s^n^)	n (−)	R^2^	Thixotropy	Area
0	53.51 ± 2.18 ^A^	0.34 ± 0.02 ^D^	0.98 ± 0.01	1157.33 ± 52.45 ^A^	26,801.67 ± 1615.65 ^A^
29	117.05 ± 2.33 ^B,C^	0.25 ± 0.02 ^A,B^	0.97 ± 0.02	3566.67 ± 76.48 ^B^	41,850.00 ± 1800.13 ^BC^
33	125.92 ± 1.79 ^E^	0.24 ± 0.04 ^A,B^	0.97 ± 0.01	4516.00 ± 124.61 ^E^	44,446.67 ± 2536.03 ^B,C^
38	139.42 ± 0.67 ^F^	0.24 ± 0.02 ^A,B^	0.98 ± 0.01	4562.83 ± 167.84 ^E^	53,178.33 ± 3242.89 ^D^
42	122.05 ± 2.19 ^C^	0.21 ± 0.03 ^A^	0.98 ± 0.01	4223.83 ± 109.58 ^D^	45,855.00 ± 1596.82 ^C^
45	120.57 ± 2.11 ^C,D^	0.23 ± 0.02 ^A,B^	0.97 ± 0.01	3980.50 ± 89.72 ^C^	42,505.00 ± 2921.69 ^B,C^
48	114.95 ± 2.83 ^B^	0.26 ± 0.01 ^C^	0.98 ± 0.01	3413.67 ± 93.66 ^B^	40,705.00 ± 2564.98 ^B^

^A–F^—Mean values between variants in the same column indicated by different letters indicate a statistically significant difference.

**Table 3 molecules-27-02397-t003:** Characteristics of açai extract and safflower oil.

	Safflower Oil	Açai Extract
TPC (mg gallic acid/g of sample)	0.27 ± 0.011	31.36 ± 1.220
ABTS (mg ascorbic acid/g of sample0	0.09 ± 0.003	50.54 ± 0.296
FRAP (mg ascorbic acid/g of sample)	0.32 ± 0.004	38.05 ± 1.268
Fatty acid profile (%)		
SFA	10.88 ± 1.36	
MUFA	9.74 ± 1.13	
PUFA	79.19 ± 0.95	
∑n-6	78.91 ± 0.98	
∑n-3	0.15 ± 0.02	

TPC—total phenolic compounds; ABTS—2,2′-azino-bis-3-ethylbenzthiazoline-6-sulphonic acid; FRAP—ferric reducing antioxidant power; SFA—saturated fatty acid; MUFA—monounsaturated fatty acid; PUFA—polyunsaturated fatty acid.

**Table 4 molecules-27-02397-t004:** Effect of fat substitute in the form of a hydrogel with encapsulated safflower oil, açai extract, and linseed flour on color parameters, browning index (BI), and pH in raw and grilled burgers at 0, 4, and 8 days of storage.

Variant	Day		Raw			
		L*	a*	b*	BI	pH
G	0	44.5 ± 2.85 ^B,C,b^	22.6 ± 2.36 ^B^	14.7 ± 1.52 ^A,B,b^	75.32 ± 6.83 ^A^	5.70 ± 0.04 ^C^
GT		46.2 ± 3.06 ^A,b^	22.7 ± 2.80 ^B^	15.2 ± 1.52 ^A,b^	74.20 ± 7.75 ^A,B^	5.71 ± 0.04 ^C^
GE		45.0 ± 3.25 ^A,B,a,b^	21.9 ± 1.93 ^B^	14.7 ± 2.01 ^A,B,b^	73.38 ± 5.89 ^A,B^	5.72 ± 0.03 ^C^
GET		43.2 ± 3.02 ^C,D,c^	22.8 ± 2.20 ^B,b^	14.0 ± 1.50 ^B,a^	75.57 ± 7.24 ^A^	5.71 ± 0.05 ^C^
CT		42.0 ± 2.66 ^D,E,b^	24.1 ± 1.81 ^A,a^	11.5 ± 0.97 ^C^	71.30 ± 7.18 ^B,a^	5.64 ± 0.04 ^A^
CO		40.4 ± 2.40 ^E,b^	22.0 ± 1.61 ^B,a^	10.4 ± 0.86 ^D^	67.10 ± 5.54 ^C,a^	5.56 ± 0.04 ^B^
G	4	44.2 ± 3.13 ^A,b^	22.3 ± 1.87 ^A^	14.7 ± 1.31 ^A,b^	75.57 ± 5.06 ^A^	5.81 ± 0.06 ^C^
GT		45.3 ± 4.24 ^A,b^	22.3 ± 2.45 ^A^	14.6 ± 1.9 ^A,b^	73.42 ± 7.15 ^A^	5.77 ± 0.03 ^C^
GE		43.7 ± 3.85 ^A,b^	21.6 ± 2.09 ^A,B^	14.0 ± 1.5 ^A,b^	73.09 ± 7.35 ^A^	5.76 ± 0.05 ^C^
GET		44.9 ± 3.78 ^A,b^	21.8 ± 2.32 ^A,B,a^	14.4 ± 1.52 ^A,a,b^	72.87 ± 8.00 ^A^	5.76 ± 0.05 ^C^
CT		41.4 ± 3.64 ^B,b^	22.8 ± 2.29 ^A,b^	11.0 ± 1.42 ^B^	68.89 ± 7.57 ^B,a^	5.63 ± 0.09 ^A^
CO		41.0 ± 3.06 ^B,b^	20.8 ± 1.68 ^B,b^	10.2 ± 1.11 ^C^	63.50 ± 6.32 ^C,b^	5.57 ± 0.07 ^B^
G	8	46.9 ± 3.16 ^a^	22.7 ± 2.18 ^A^	16.3 ± 1.51 ^A,a^	76.35 ± 6.28 ^A^	5.73 ± 0.04 ^B^
GT		48.6 ± 3.55 ^a^	21.1 ± 2.47 ^A^	16.6 ± 1.45 ^A,a^	72.36 ± 6.68 ^A^	5.73 ± 0.03 ^B^
GE		46.9 ± 3.02 ^a^	21.1 ± 1.15 ^A^	16.0 ± 1.51 ^A,a^	73.44 ± 5.66 ^A^	5.75 ± 0.01 ^B^
GET		47.5 ± 4.00 ^a^	20.9 ± 2.54 ^A,a,c^	15.4 ± 1.69 ^A,b^	70.18 ± 8.77 ^A^	5.75 ± 0.03 ^B^
CT		46.9 ± 6.54 ^a^	18.5 ± 2.94 ^B,b^	10.9 ± 1.76 ^B^	55.01 ± 11.42 ^B,b^	5.58 ± 0.03 ^A^
CO		44.9 ± 5.44 ^a^	18.6 ± 2.03 ^B,c^	10.8 ± 1.51 ^B^	56.83 ± 7.73 ^B,c^	5.58 ± 0.03 ^A^
		**Grilled**				
		**L***	**a***	**b***	**BI**	
G	0	52.2 ± 1.68 ^A^	7.4 ± 0.75 ^A,B,a,b^	13.5 ± 0395 ^A^	39.87 ± 2.66 ^A,b^	
GT		52.7 ± 1.71 ^A^	7.0 ± 0.40 ^A,a^	13.7 ± 1.25 ^A^	39.27 ± 2.38 ^A^	
GE		51.4 ± 1.92 ^A,a^	7.3 ± 0.81 ^A,B,a^	12.6 ± 1.47 ^A,a^	38.09 ± 2.95 ^A,b^	
GET		52.7 ± 1.38 ^A,a^	7.3 ± 0.56 ^A,B,a^	13.5 ± 1.32 ^A^	39.27 ± 2.80 ^A,b^	
CT		51.6 ± 2.25 ^A^	7.7 ± 1.00 ^B,a^	10.0 ± 0.88 ^B,a^	32.18 ± 1.75 ^B,a^	
CO		49.1 ± 2.37 ^B,a^	7.9 ± 0.66 ^B^	9.8 ± 0.62 ^B,a^	33.72 ± 2.53 ^B,a^	
G	4	52.5 ± 1.61	7.0 ± 0.41 ^A,C,b^	13.0 ± 0.84 ^A^	37.83 ± 1.69 ^B,b^	
GT		52.3 ± 2.378	7.3 ± 0.45 ^A,b^	13.7 ± 1.30 ^A^	40.03 ± 2.32 ^A^	
GE		53.9 ± 1.81 ^b^	6.7 ± 0.35 ^C,D,b,c^	13.7 ± 1.16 ^A,b^	37.96 ± 2.87 ^B,b^	
GET		53.3 ± 1.85 ^a^	6.2 ± 0.50 ^D,b^	13.9 ± 0.62 ^A^	38.43 ± 2.16 ^A,B,b^	
CT		53.1 ± 2.04	7.1 ± 0.45 ^A,C,b^	9.6 ± 1.04 ^B,a^	29.23 ± 1.94 ^D,b^	
CO		51.0 ± 1.44 ^b^	7.8 ± 0.23 ^B^	9.4 ± 0.55 ^B,b,c^	31.21 ± 1.80 ^C,b^	
G	8	51.4 ± 4.93 ^A^	10.7 ± 5.29 ^A,a^	12.7 ± 1.73 ^A^	43.74 ± 9.60 ^A,a^	
GT		52.7 ± 2.38 ^A^	7.7 ± 0.72 ^A,b^	13.6 ± 1.10 ^A^	40.02 ± 2.54 ^B^	
GE		48.6 ± 4.44 ^B,C,a^	6.8 ± 0.59 ^B,a,c^	13.5 ± 1.14 ^A,b^	42.62 ± 4.18 ^A,B,a^	
GET		51.0 ± 2.12 ^A,C,b^	7.0 ± 0.56 ^B,a^	13.7 ± 0.85 ^A^	41.03 ± 2.60 ^A,B,a^	
CT		52.2 ± 1.40 ^A^	8.1 ± 0.63 ^A,a^	9.0 ± 0.80 ^B,b^	29.84 ± 2.10 ^D,b^	
CO		47.6 ± 1.64 ^B,c^	8.0 ± 0.37 ^A^	9.6 ± 0.70 ^B,a,c^	34.54 ± 1.97 ^C,a^	

^A–E^ Mean values between variants on the same storage day with different letters indicate a significant difference. ^a–c^ Mean values of the same variants between storage day with different letters indicate a significant difference. G—encapsulated oil; GT—encapsulated oil + tallow; GE—encapsulated oil with açai extract; GET—encapsulated oil with açai extract + tallow; CT—control with tallow; CO—control with oil. L*—lightness; a*—redness; b*—yellowness; BI—browning index.

**Table 5 molecules-27-02397-t005:** Effect of a hydrogel emulsion with capsulated safflower oil, açai extract, and linseed flour as a fat substitute, on texture parameters (TPA) of grilled burgers at 0, 4, and 8 days of storage.

Variant	Springiness (−)	Chewiness (N)	Cohesiveness (−)	Hardness (N)
G	0.4 ± 0.20	0.6 ± 0.26 ^B,a,b^	0.1 ± 0.04 ^C,a^	6.4 ± 1.59 ^B^
GT	0.4 ± 0.11 ^a^	0.3 ± 0.17 ^B^	0.0 ± 0.05 ^B,C^	5.3 ± 1.23 ^B^
GE	0.8 ± 0.30 ^a^	0.4 ± 0.26 ^B^	0.1 ± 0.04 ^B,C^	5.5 ± 0.98 ^B^
GET	0.5 ± 0.2	0.5 ± 0.24 ^B^	0.1 ± 0.10 ^B,C^	5.6 ± 1.53 ^B^
CT	0.9 ± 0.22	10.6 ± 2.30 ^A,b^	0.4 ± 0.03 ^A^	27.5 ± 5.25 ^A,b^
CO	0.5 ± 0.12 ^a^	8.8 ± 1.50 ^A,c^	0.4 ± 0.04 ^A^	24.8 ± 1.95 ^A,c^
G	0.2 ± 0.05 ^C^	1.0 ± 0.49 ^B,a^	0.1 ± 0.03 ^C,a^	9.5 ± 2.16 ^B^
GT	0.2 ± 0.03 ^C,b^	0.4 ± 0.59 ^B^	0.0 ± 0.07 ^B,C^	7.8 ± 2.68 ^B^
GE	0.1 ± 0.03 ^C,b^	0.5 ± 0.62 ^B^	0.0 ± 0.07 ^B,C^	7.9 ± 2.46 ^B^
GET	0.1 ± 0.04 ^C^	0.0 ± 0.36 ^B^	0.0 ± 0.09 ^B,C^	5.8 ± 1.20 ^B^
CT	0.4 ± 0.06 ^A^	17.1 ± 4.73 ^A,a,b^	0.4 ± 0.03 ^A^	47.3 ± 12.23 ^A,a,b^
CO	0.3 ± 0.04 ^B,b^	21.4 ± 4.25 ^A,b^	0.4 ± 0.02 ^A^	58.6 ± 12.04 ^A,b^
G	0.1 ± 0.03 ^C^	−0.3 ± 0.90 ^C,b^	−0.1 ± 0.14 ^B,b^	10.4 ± 3.34 ^C^
GT	0.1 ± 0.03 ^C,b^	−0.5 ± 0.70 ^C^	−0.1 ± 0.08 ^B^	10.2 ± 2.51 ^C^
GE	0.2 ± 0.04 ^C,b^	−0.1 ± 0.99 ^C^	0.0 ± 0.10 ^B^	11.6 ± 3.93 ^C^
GET	0.1 ± 0.03 ^C^	0.5 ± 1.43 ^C^	0.0 ± 0.12 ^B^	15.0 ± 7.16 ^C^
CT	0.5 ± 0.04 ^A^	19.6 ± 6.80 ^B,a^	0.3 ± 0.04 ^A^	58.6 ± 17.54 ^B,a^
CO	0.3 ± 0.01 ^B,b^	29.1 ± 6.22 ^A,a^	0.3 ± 0.03 ^A^	90.9 ± 18.76 ^A,a^

^A–C^ Mean values between variants on the same storage day with different letters indicate a significant difference. ^a–c^ Mean values of the same variants between storage day with different letters indicate a significant difference. G—encapsulated oil; GT—encapsulated oil + tallow; GE—encapsulated oil with açai extract; GET—encapsulated oil with açai extract + tallow; CT—control with tallow; CO—control with oil.

**Table 6 molecules-27-02397-t006:** Effects of a fat substitute on the volatile compounds profile in raw and grilled burger after 0, 4, and 8 days of storage in cold conditions.

Raw Burger																		
Volatile Compounds	0						4						8					
	G	GT	GE	GET	CT	CO	G	GT	GE	GET	CT	CO	G	GT	GE	GET	CT	CO
aldehyde																		
(E,E) −2,4−hexadienal							+	+					+	+				
2−-Methylpropanal	+	+	+	+	+	+	+	+	+	+	+		+	+	+	+		
3−Methylbutanal												+					+	+
Benzaldehyde																		+
Benzeneacetaldehyde							+	+	+	+			+	+	+	+		
Propanal	+	+	+	+	+	+	+	+	+	+	+	+	+	+	+	+	+	+
alcohol																		
1−Hexanol							+	+	+	+	+	+	+	+	+	+	+	+
1−Hexen−3−ol									+						+			
1−Penten−3−ol																+		
1−Propanol	+	+	+	+	+	+	+	+	+	+	+	+	+	+	+	+	+	+
1−Propanol, 2−methyl	+	+	+				+	+	+	+			+	+	+	+		
2−Nonen−1−ol									+						+		+	+
Pentan−2−ol								+		+				+			+	+
ester																		
Ethyl 2−methylbutyrate										+						+		
Ethyl isobutyrate	+	+	+		+	+				+						+		
Hexyl propionate												+						
Methyl isobutyrate	+	+	+		+	+												
Propyl propanoate								+	+	+		+		+	+	+	+	+
ketone																		
1−Hexen−3−one										+						+		
2,3−Butanediol																	+	+
Sotolon									+					+	+			
acid																		
Acetic acid	+	+	+	+	+	+	+	+	+	+	+	+		+	+	+	+	+
Pentanoic acid										+		+				+		
Propanoic acid		+	+					+	+					+	+			
acetate																		
Bezyl acetate							+					+	+					
Isoamyl acetate																	+	
Isopropyl acetate	+	+	+	+	+	+												
terpene																		
Alpha−phellandrene												+						
Alphapinene			+	+		+			+	+		+			+			+
Limonene				+						+								
sulphur compounds																		
Dimethyl sulfide																	+	+
2−Methyl−2−propanethiol							+	+	+	+	+		+	+	+	+		
**Grilled Burger**																		
Volatile compounds	0						4						8					
	G	GT	GE	GET	CT	CO	G	GT	GE	GET	CT	CO	G	GT	GE	GET	CT	CO
aldehyde																		
(E) −3−hexenal	+	+					+	+			+		+	+				
(E,E) −2,4−hexadienal	+	+					+	+	+				+	+	+			
2−Methylpropanal			+	+	+						+		+		+			
3−Methylbutanal					+													
Benzaldehyde		+						+						+			+	
Benzeneacetaldehyde	+	+	+	+		+	+	+	+	+		+	+	+	+	+		+
Methional						+				+		+						+
Propanal						+		+				+			+			+
alcohol																		
1−Hexanol										+	+					+	+	
1−Hexen−3−ol			+	+		+			+	+		+			+			+
1−Propanol	+	+	+	+	+	+	+	+	+	+	+	+	+	+	+	+	+	+
1−Propanol, 2-methyl	+	+	+	+			+	+	+	+			+	+	+	+		
2−Furanmethanol						+						+						+
n−Butanol																	+	
Pentan−2−ol			+	+					+	+					+			
esther																		
ethyl 2−methylbutyrate				+						+								
propyl propanoate																	+	
ketone																		
1−Hexen−3−one																+		
2,3-Butanediol																		+
2−Acetyl−1−pyrroline					+						+						+	
Acetophenone					+	+						+						+
acid																		
3−Methylbutanoic acid	+						+						+					
Acetic acid		+	+	+	+			+		+							+	
Benzoic acid																		+
acetate																		
Ethyl acetate	+	+	+	+		+		+	+	+				+	+	+		
terpene																		
Alpha−phellandrene																	+	
Alphapinene			+	+		+			+	+		+			+	+		+
nitrogenous compounds																		
Pyrazine	+	+	+	+			+	+	+	+			+	+	+	+	+	
Pyrrole					+		+						+	+			+	
sulphur compounds																		
2−Methyl−2−propanethiol									+	+	+		+	+				
Dimethyl sulfide					+						+						+	

G—encapsulated oil; GT—encapsulated oil + tallow; GE—encapsulated oil with açai extract; GET—encapsulated oil with açai extract + tallow; CT—control with tallow; CO—control with oil.

**Table 7 molecules-27-02397-t007:** Effects of a fat substitute (hydrogel emulsion with encapsulated oil and acai extract) on the fatty acid profile of raw and grilled burgers at 0, 4, and 8 days of storage in cold conditions.

Raw Burger						
	G	GT	GE	GET	CT	CO
**SFA**						
0	38.2 ± 0.46 ^A^	36.3 ± 2.62 ^A,B^	35.0 ± 0.51 ^B^	38.9 ± 1.00 ^A^	39.4 ± 0.37 ^A^	32.7 ± 0.18 ^C^
4	41.5 ± 0.16 ^B^	40.7 ± 0.52 ^B^	41.6 ± 1.61 ^B^	42.7 ± 0.62 ^B^	50.1 ± 0.03 ^A^	37.7 ± 0.29 ^C^
8	43.8 ± 0.60 ^B^	45.3 ± 1.08 ^B^	40.3 ± 0.48 ^C^	44.1 ± 1.65 ^B^	50.5 ± 0.64 ^A^	51.6 ± 2.13 ^A^
**MUFA**						
0	35.9 ± 0.26 ^D^	43.4 ± 2.47 ^B^	39.7 ± 0.32 ^B^	41.2 ± 0.92 ^B^	58.8 ± 0.45 ^A^	37.5 ± 0.42 ^C^
4	36.7 ± 1.26 ^C^	39.9 ± 0.26 ^B^	32.5 ± 0.52 ^E^	38.2 ± 0.27 ^C^	48.5 ± 0.01 ^A^	34.2 ± 0.79 ^D^
8	33.6 ± 1.09 ^C^	36.5 ± 0.51 ^B^	36.7 ± 0.55 ^B^	37.1 ± 1.32 ^B^	48.0 ± 0.68 ^A^	28.9 ± 1.70 ^D^
**PUFA**						
0	25.4 ± 0.78 ^B^	19.7 ± 0.12 ^C^	24.8 ± 0.14 ^B^	19.5 ± 0.07 ^C^	1.0 ± 0.02 ^D^	28.8 ± 0.10 ^A^
4	21.4 ± 1.39 ^C^	18.9 ± 0.75 ^D^	24.0 ± 1.09 ^B^	18.7 ± 0.67 ^D^	0.9 ± 0.01 ^E^	27.6 ± 0.51 ^A^
8	20.2 ± 1.69 ^B^	17.7 ± 0.58 ^C^	22.6 ± 0.10 ^A^	18.3 ± 0.21 ^C^	1.0 ± 0.02 ^D^	19.1 ± 0.44 ^B^
**CLA**						
0	0.46 ± 0.06 ^C^	0.57 ± 0.04 ^B^	0.50 ± 0.05 ^C^	0.49 ± 0.01 ^C^	0.72 ± 0.06 ^A^	0.49 ± 0.00 ^C^
4	0.50 ± 0.02 ^A^	0.48 ± 0.04 ^B^	0.38 ± 0.00 ^C^	0.45 ± 0.00 ^B^	0.54 ± 0.01 ^A^	0.40 ± 0.02 ^C^
8	0.45 ± 0.00 ^B^	0.41 ± 0.01 ^B^	0.41 ± 0.03 ^B^	0.47 ± 0.06 ^A,B^	0.54 ± 0.02 ^A^	0.40 ± 0.01 ^B^
**∑** **n3**						
0	3.24 ± 0.46 ^B^	3.16 ± 0.06 ^B^	3.95 ± 0.16 ^A^	3.25 ± 0.01 ^B^	0.52 ± 0.01 ^C^	0.58 ± 0.04 ^C^
4	3.15 ± 0.69 ^A,B^	2.69 ± 0.46 ^B^	3.91 ± 0.16 ^A^	3.12 ± 0.05 ^B^	0.46 ± 0.01 ^D^	0.53 ± 0.01 ^C^
8	3.33 ± 0.72 ^A^	2.18 ± 0.07 ^B^	3.53 ± 0.05 ^A^	3.15 ± 0.27 ^A^	0.60 ± 0.07 ^C^	0.46 ± 0.05 ^D^
**∑** **n6**						
0	21.98 ± 0.31 ^B^	16.35 ± 0.11 ^C^	20.65 ± 0.02 ^B^	16.00 ± 0.06 ^C^	0.31 ± 0.02 ^D^	28.01 ± 0.06 ^A^
4	18.01 ± 0.69 ^C^	16.04 ± 0.28 ^D^	19.88 ± 0.92 ^B^	15.40 ± 0.62 ^D^	0.30 ± 0.00 ^E^	26.88 ± 0.50 ^A^
8	18.73 ± 0.96 ^A^	15.39 ± 0.51 ^B^	18.86 ± 0.06 ^A^	16.48 ± 1.44 ^A,B^	0.25 ± 0.05 ^C^	18.45 ± 0.51 ^A^
**∑** **n6/** **∑** **n3**						
0	6.92 ± 0.89 ^B^	5.18 ± 0.03 ^C^	5.24 ± 0.22 ^C^	4.92 ± 0.01 ^C^	0.60 ± 0.04 ^D^	48.29 ± 3.04 ^A^
4	5.95 ± 1.08 ^B^	6.12 ± 0.93 ^B^	5.09 ± 0.03 ^C^	4.94 ± 0.12 ^C^	0.66 ± 0.02 ^D^	50.57 ± 0.35 ^A^
8	5.84 ± 0.98 ^C^	7.06 ± 0.01 ^B^	5.35 ± 0.05 ^C^	5.23 ± 0.00 ^C^	0.44 ± 0.13 ^D^	41.22 ± 5.84 ^A^
**Tl**						
0	0.95 ± 0.05 ^B^	0.88 ± 0.09 ^B,C^	0.80 ± 0.02 ^C^	0.97 ± 0.04 ^B^	1.16 ± 0.02 ^A^	0.91 ± 0.00 ^B^
4	1.08 ± 0.06 ^B^	1.09 ± 0.06 ^B^	1.10 ± 0.08 ^B^	1.13 ± 0.02 ^B^	1.78 ± 0.01 ^A^	1.13 ± 0.01 ^B^
8	1.17 ± 0.09 ^D^	1.34 ± 0.06 ^C^	1.01 ± 0.02 ^D^	1.17 ± 0.01 ^D^	1.72 ± 0.02 ^B^	1.99 ± 0.15 ^A^
**Al**						
0	0.49 ± 0.01 ^B^	0.54 ± 0.04 ^B^	0.49 ± 0.01 ^B^	0.55 ± 0.01 ^B^	0.69 ± 0.04 ^A^	0.44 ± 0.03 ^C^
4	0.58 ± 0.01 ^B^	0.57 ± 0.01 ^B^	0.52 ± 0.02 ^B^	0.60 ± 0.00 ^B^	0.85 ± 0.00 ^A^	0.49 ± 0.00 ^B^
8	0.59 ± 0.02 ^CD^	0.65 ± 0.03 ^C^	0.54 ± 0.00 ^D^	0.61 ± 0.02 ^C^	0.90 ± 0.02 ^A^	0.72 ± 0.09 ^B^
**h/H**						
0	2.53 ± 0.01 ^B^	2.37 ± 0.21 ^B,C^	2.59 ± 0.02 ^B^	2.21 ± 0.06 ^C^	1.77 ± 0.08 ^C^	2.93 ± 0.03 ^A^
4	2.14 ± 0.02 ^B^	2.11 ± 0.02 ^B^	2.29 ± 0.10 ^B^	1.99 ± 0.01 ^C^	1.33 ± 0.01 ^D^	2.50 ± 0.01 ^A^
8	2.06 ± 0.10 ^B^	1.89 ± 0.08 ^C^	2.24 ± 0.03 ^A^	1.91 ± 0.02 ^C^	1.25 ± 0.02 ^D^	1.63 ± 0.19 ^C^
	**Grilled Burger**					
	**G**	**GT**	**GE**	**GET**	**CT**	**CO**
**SFA**						
0	42.93 ± 0.69 ^A^	40.86 ± 2.47 ^A^	50.20 ± 1.25 ^A^	48.88 ± 4.36 ^A^	50.21 ± 1.06 ^A^	47.47 ± 10.32 ^A^
4	43.81 ± 1.28 ^A,B,C^	41.06 ± 1.86 ^B,C^	38.56 ± 1.89 ^C^	45.35 ± 0.83 ^A,B^	50.21 ± 1.67 ^A^	42.85 ± 1.84 ^B,C^
8	43.92 ± 0.71 ^A^	40.86 ± 2.47 ^A^	51.66 ± 0.81 ^A^	47.48 ± 2.38 ^A^	50.04 ± 0.84 ^A^	51.41 ± 15.90 ^A^
**MUFA**						
0	34.06 ± 3.32 ^B,C^	38.54 ± 0.57 ^A,B^	31.52 ± 2.71 ^B,C^	32.32 ± 0.27 ^B,C^	46.85 ± 1.32 ^A^	28.64 ± 2.58 ^C^
4	34.94 ± 0.11 ^B^	39.55 ± 0.90 ^B^	37.00 ± 1.74 ^B^	35.81 ± 1.53 ^B^	47.95 ± 0.74 ^A^	40.66 ± 2.50 ^B^
8	35.33 ± 1.50 ^B,C^	38.34 ± 0.85 ^B^	32.73 ± 0.99 ^C,D^	32.32 ± 0.27 ^C,D^	47.36 ± 0.60 ^A^	28.64 ± 2.58 ^D^
**PUFA**						
0	20.55 ± 0.53 ^A^	19.40 ± 0.36 ^A^	19.86 ± 1.49 ^A^	18.73 ± 4.20 ^A^	2.76 ± 0.00 ^B^	12.65 ± 3.00 ^A^
4	21.25 ± 1.39 ^B^	18.46 ± 0.36 ^B,C^	27.01 ± 0.00 ^A^	17.89 ± 1.01 ^C^	3.91 ± 0, 03 ^D^	18.27 ± 0.22 ^C^
8	20.75 ± 0.79 ^A^	19.40 ± 0.36 ^A^	20.11 ± 1.14 ^A^	20.20 ± 2.11 ^A^	2.77 ± 0.00 ^C^	12.65 ± 3.00 ^B^
**CLA**						
0	0.21 ± 0.04 ^B^	0.26 ± 0.02 ^A,B^	0.26 ± 0.02 ^A,B^	0.20 ± 0.01 ^B^	0.33 ± 0.03 ^A^	0.23 ± 0.04 ^A,B^
4	0.25 ± 0.01 ^A^	0.27 ± 0.03 ^A^	0.30 ± 0.03 ^A^	0.24 ± 0.01 ^A^	0.32 ± 0.02 ^A^	0.25 ± 0.04 ^A^
8	0.21 ± 0.04 ^B^	0.26 ± 0.02 ^A,B^	0.25 ± 0.02 ^A,B^	0.20 ± 0.01 ^B^	0.33 ± 0.03 ^A^	0.23 ± 0.04 ^A,B^
**∑** **n3**						
0	3.56 ± 0.13 ^A^	2.80 ± 0.61 ^A^	3.28 ± 0.21 ^A^	3.45 ± 0.37 ^A^	0.46 ± 0.03 ^B^	0.61 ± 0.27 ^B^
4	3.40 ± 0.41 ^A,B^	2.51 ± 0.01 ^B^	4.26 ± 0.00 ^A^	3.95 ± 0.53 ^A^	0.52 ± 0.08 ^C^	0.74 ± 0.03 ^C^
8	3.03 ± 0.63 ^A^	3.01 ± 0.32 ^A^	3.28 ± 0.21 ^A^	3.45 ± 0.37 ^A^	0.46 ± 0.03 ^B^	0.48 ± 0.09 ^B^
**∑** **n6**						
0	16.34 ± 0.49 ^A^	15.96 ± 0.28 ^A^	15.31 ± 0.38 ^A^	14.89 ± 3.55 ^A^	1.76 ± 0.24 ^B^	11.59 ± 2.83 ^A^
4	19.28 ± 4.15 ^A,B^	15.29 ± 0.35 ^A,B^	22.05 ± 0.00 ^A^	13.32 ± 0.52 ^B^	2.49 ± 0.38 ^C^	16.78 ± 0.10 ^A,B^
8	17.07 ± 1.51 ^A^	17.53 ± 2.51 ^A^	16.02 ± 1.27 ^A^	16.19 ± 1.71 ^A^	1.59 ± 0.01 ^C^	11.59 ± 2.83 ^B^
**∑** **n6/** **∑** **n3**						
0	4.59 ± 0.03 ^C^	6.86 ± 0.09 ^B^	4.96 ± 0.03 ^B,C^	4.61 ± 0.12 ^C^	3.77 ± 0.27 ^C^	24.08 ± 1.33 ^A^
4	5.48 ± 0.76 ^B,C^	6.10 ± 0.11 ^B^	5.18 ± 0.00 ^B,C,D^	3.39 ± 0.32 ^D^	3.70 ± 0.20 ^C,D^	22.55 ± 0.88 ^A^
8	5.82 ± 1.71 ^B^	5.90 ± 1.45 ^B^	4.96 ± 0.02 ^B^	4.64 ± 0.07 ^B^	3.60 ± 0.02 ^B^	24.08 ± 1.33 ^A^
**Tl**						
0	1.19 ± 0.08 ^A^	1.09 ± 0.11 ^A^	1.38 ± 0.21 ^A^	1.42 ± 0.29 ^A^	1.90 ± 0.03 ^A^	2.16 ± 0.33 ^A^
4	1.02 ± 0.26 ^B^	1.15 ± 0.02 ^B^	0.85 ± 0.00 ^B^	1.18 ± 0.08 ^B^	1.87 ± 0.14 ^A^	1.37 ± 0.11 ^A,B^
8	1.19 ± 0.09 ^A^	1.14 ± 0.03 ^A^	1.60 ± 0.52 ^A^	1.32 ± 0.15 ^A^	1.87 ± 0.07 ^A^	2.16 ± 0.33 ^A^
**Al**						
0	0.63 ± 0.01 ^A^	0.58 ± 0.03 ^A^	0.63 ± 0.02 ^A^	0.70 ± 0.17 ^A^	0.85 ± 0.01 ^A^	0.94 ± 0.14 ^A^
4	0.54 ± 0.11 ^A,B^	0.57 ± 0.04 ^A,B^	0.50 ± 0.00 ^B^	0.65 ± 0.09 ^A,B^	0.76 ± 0.03 ^A^	0.57 ± 0.00 ^A,B^
8	0.62 ± 0.02 ^A^	0.56 ± 0.05 ^A^	0.69 ± 0.10 ^A^	0.58 ± 0.05 ^A^	0.84 ± 0.02 ^A^	0.79 ± 0.35 ^A^
**h/H**						
0	1.93 ± 0.04 ^A^	2.09 ± 0.12 ^A^	1.91 ± 0.06 ^A^	1.98 ± 0.11 ^A^	1.32 ± 0.00 ^B^	1.21 ± 0.20 ^B^
4	2.38 ± 0.35 ^A^	2.17 ± 0.15 ^A,B^	2.50 ± 0.00 ^A^	1.82 ± 0.30 ^A,B^	1.42 ± 0.20 ^B^	2.11 ± 0.01 ^A,B^
8	2.00 ± 0.05 ^A^	2.17 ± 0.23 ^A^	1.85 ± 0.15 ^A,B^	1.98 ± 0.11 ^A^	1.34 ± 0.02 ^B,C^	1.21 ± 0.20 ^C^

^A–E^ Mean values between variants on the same storage day with different letters indicate a significant difference. SFA—saturated fatty acid; MUFA—monounsaturated fatty acid; PUFA—polyunsaturated fatty acid; CLA—conjugated linoleic acid; Al—atherogenic index; Tl—thrombogenicity index; h/H—hypocholesterolemic/hypercholesterolemic ratio; G—encapsulated oil; GT—encapsulated oil + tallow; GE—encapsulated oil with açai extract; GET—encapsulated oil with açai extract + tallow; CT—control with tallow, CO—control with oil.

**Table 8 molecules-27-02397-t008:** Analysis of correlations among fatty acid profile (SFA, MUFA, and PUFA), color parameters (L*, a*, b*, and BI) and WHC, pH, weight and cooking loss, TBARS, and TPA parameters (springiness, chewiness, cohesiveness, and hardness) with raw and/or grilled burgers.

Raw Burger												
	SFA	MUFA	PUFA	L*	a*	b*	BI	WHC	pH	Weight Loss	TBARS	
**SFA**	1	−0.040	−0.580 *	0.345	−0.523 *	−0.116	−0.474 *	−0.292	−0.108	0.677 *	0.445	
**MUFA**		1	−0.790 *	−0.230	0.396	−0.282	−0.027	−0.309	−0.201	−0.311	−0.319	
**PUFA**			1	−0.020	−0.001	0.307	0.318	0.438	0.236	−0.162	−0.006	
**L***				1	−0.343	0.727 *	0.112	0.539 *	0.497 *	0.008	0.848 *	
**a***					1	0.277	0.809 *	0.345	0.381	−0.750 *	−0.226	
**b***						1	0.751 *	0.928 *	0.880 *	−0.473 *	0.765 *	
**BI**							1	0.810 *	0.797 *	−0.764 *	0.281	
**WHC**								1	0.925 *	−0.588 *	0.654 *	
**pH**									1	−0.487 *	0.698 *	
**Weight loss**										1	0.072	
**TBARS**											1	
**Grilled burger**												
	**SFA**	**MUFA**	**PUFA**	**Springiness**	**Chewiness**	**Cohesiveness**	**Hardness**	**Cooking loss**	**L***	**a***	**b***	**BI**
**SFA**	1	−0.040	−0.611 *	0.401	0.435	0.416	0.440	0.395	−0.546 *	0.035	−0.492 *	−0.345
**MUFA**		1	−0.612 *	0.129	0.285	0.391	0.210	0.306	0.506 *	−0.004	−0.384	−0.565 *
**PUFA**			1	−0.408	−0.657 *	−0.765 *	−0.594 *	−0.739 *	0.133	−0.125	0.790 *	0.782 *
**Springiness**				1	0.139	0.437	0.031	0.176	−0.026	−0.007	−0.329	−0.367
**Chewiness**					1	0.850 *	0.985 *	0.891 *	−0.428	0.185	−0.905 *	−0.811 *
**Cohesiveness**						1	0.759 *	0.901 *	−0.268	0.060	−0.936 *	−0.937 *
**Hardness**							1	0.845 *	−0.492 *	0.220	−0.848 *	−0.716 *
**Cooking loss**								1	−0.433	0.181	−0.920 *	−0.830 *
**L***									1	−0.267	0.382	0.024
**a***										1	−0.302	0.076
**b***											1	0.887 *
**BI**												1

*—*p* < 0.05.

**Table 9 molecules-27-02397-t009:** Beef burger composition.

Variant	Component			
	Type of Fat Added	Linseed Flour (g)	Meat (g)	Salt (g)
G	9.5 g Hydrogel with encapsulated oil	10.5	100	1.7
GT	9.5 g Hydrogel with encapsulated oil + 5 g beef tallow	10.5	95	1.7
GE	9.5 g Hydrogel with encapsulated oil with açai extract	10.5	100	1.7
GET	9.5 g Hydrogel with encapsulated oil with açai extract + 5 g beef tallow	10.5	95	1.7
CT	20 g Beef tallow	-	100	1.7
CO	8 g Oil	-	112	1.7

G—encapsulated oil; GT—encapsulated oil + tallow; GE—encapsulated oil with açai extract; GET—encapsulated oil with açai extract + tallow; CT—control with tallow; CO—control with oil.

## Data Availability

Available from the authors.

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
