# Peer review of "Hydrogel Emulsion with Encapsulated Safflower Oil Enriched with Açai Extract as a Novel Fat Substitute in Beef Burgers Subjected to Storage in Cold Conditions"

_molecules, 2022, doi:10.3390/molecules27082397_

Round 1
Reviewer 1 Report
Authors describes the Hydrogel emulsion with encapsulated safflower oil enriched with açai extract as a novel fat substitute in beef burgers subjected to storage in cold conditions. The topic is of a certain interest however some corrections and additional experiments are necessary before publication:
- Abstract is too descriptive. Please add the main obtained results;
- Clarify better the aim of the work;
- Calculate the browning index using CIELab parameters;
- Why authors did not perform microbial analysis at the end of storage? Please add.
- There is a correlation between fatty acids composition and storage?
- Why volatiles are not quantified?
- DPPH and ABTS are inconsistent in this study. Why authors perform the radical scavenging evaluation of these samples?
- Sensorial analysis is necessary in order to clarify the acceptability of this as fat replacer. Please add.
- Conclusion should be rewritten in order to highlight the novelty of the proposed study.
Reviewer 2 Report
This study focuses on the hydrogel emulsion with encapsulated safflower oil enriched with açai extract as a novel fat substitute in beef burgers subjected to storage in cold conditions. I have thoroughly reviewed the manuscript. The subject seems to be important and a lot of workload has been done. However, there are many uncertainties in the realization of the research hypothesis. It is not clear how the conditions such as used oil ratios, storage temperatures are determined, and the parameters taken as criteria in the samples selected as a result of preliminary trials. In addition, although many analyzes have been made, the results of the analysis have not been discussed at all, especially for some parameters. Considering the number of analyzes, it was expected that a correlation study would be conducted to understand the relationship between parameters. Principal Component Analysis would be an excellent way to show these findings on the same graph (biplot). Method details in sections 3.1 and 3.2 are not detailed enough to produce reproducible analysis. There are also some errors in various parts of the manuscript, indicating that it was not written very carefully. For example, line 441, 449, 469, etc.
Reviewer 3 Report
This manuscript focused on the development of a novel fat substitute in beef burgers based on safflower oil and acai extract. The utilization of this fat substitute contributed to the formulation of a healthier meat product. The experimental design was appropriate, and the results were interesting. Therefore, I suggest a minor revision of this manuscript. Detailed comments were as below:
(1) Grammar check is strongly recommended.
(2) Lines 25-26: Did you determine the shelf life of beef burgers?
(3) Lines 77-79: The safety status of these chemical compounds should be provided.
(4) Line 78-79: What were the advantages of the combination of safflower oil and acai extract?
(5) Please discuss the correlation between different parameters of beef burgers.
(6) Line 143: What was the exact concentration of encapsulated oil in hydrogels presented in Fig. 2C and Fig. 2D?
(7) Line 400: Change “Material” to “Materials”. In addition, more details regarding the methods should be provided.
Reviewer 4 Report
It is really a very good work. The design is correct, many methods and techniques were used, and the results are clear and promising. I suggest the publication of this ms.
Round 2
Reviewer 1 Report
Unfortunately, sensory analysis was not performed which is a crucial point in the development of a new recipe. Therefore I would suggest performing these analyzes and submitting this article again when all results have been obtained.
Reviewer 2 Report
I recommend for accept.